# Is the CYP2D6 Genotype Associated with Antipsychotic-Induced Weight Gain? [note 1]

**DOI:** 10.3390/jpm12101728

**Published:** 2022-10-17

**Authors:** Gesche Jürgens, Benjamin Skov Kaas-Hansen, Merete Nordentoft, Thomas Werge, Stig Ejdrup Andersen

**Affiliations:** 1Clinical Pharmacology Unit, Zealand University Hospital, Sygehusvej 10, 4000 Roskilde, Denmark; 2Department of Clinical Medicine, University of Copenhagen, 2200 Copenhagen, Denmark; 3Department of Intensive Care, Copenhagen University Hospital—Rigshospitalet, 2100 Copenhagen, Denmark; 4Section of Biostatistics, Department of Public Health, University of Copenhagen, 1353 Copenhagen, Denmark; 5Copenhagen Research Center for Mental Health-CORE, 2900 Hellerup, Denmark; 6Mental Health Centre Sct. Hans, 4000 Roskilde, Denmark

**Keywords:** antipsychotic-induced weight gain, schizophrenia, pharmacogenetics, CYP2D6 polymorphism

## Abstract

Antipsychotic-induced weight gain (AIWG) is a serious adverse effect. Studies have linked genetically-predicted CYP2D6 metabolic capacity to AIWG. The evidence, however, is ambiguous. We performed multiple regression analyses examining the association between genetic-predicted CYP2D6 metabolic capacity and AIWG. Analyses were based on previously unpublished data from an RCT investigating the clinical utility of routine genotyping of CYP2D6 and CYP2C19 in patients with schizophrenia. A total of 211 patients, corresponding to 71% of the original study population, were included. Our analyses indicated an effect of genetically predicted CYP2D6 metabolic capacity on AIWG with significant weight gain in both CYP2D6 poor metabolizers (PMs) (4.00 kg (95% CI: 0.80; 7.21)) and ultrarapid metabolizers (UMs) (6.50 kg (95% CI: 1.03; 12.0)). This finding remained stable after adjustment for covariates (PMs: 4.26 kg (0.88; 7.64), UMs: 7.26 kg (1.24; 13.3)). In addition to the CYP2D6 metabolic capacity, both baseline body mass index (−0.24 (95% CI: −0.44; −0.03)) and chlorpromazine equivalents per day (0.0041 (95% CI: 0.0005; 0.0077)) were statistically significantly associated with weight change in the adjusted analysis. Our results support that the genetically predicted CYP2D6 metabolic capacity matters for AIWG.

## 1. Introduction

Antipsychotics are routinely used to treat schizophrenia. Their effects are well documented, and for many patients, antipsychotics are indispensable for alleviating or relieving psychotic symptoms. Side effects are common, however, with some being severe and disabling.

In particular, second-generation antipsychotics (SGAs) have been associated with metabolic disturbances, including weight gain, not only affecting patients’ quality of life and adherence to drug treatment but also increasing the risk of comorbidities associated with increased mortality such as diabetes, dyslipidemia and ischemic heart disease [1,2,3].

Although research supports that the use of antipsychotics compared to non-use reduces all cause-mortality in patients suffering from schizophrenia, it is important to keep in mind that metabolic complications contribute considerably to the excess mortality in schizophrenia [4,5,6]. Thus, clinical monitoring of antipsychotic treatment requires timely adjustments that adapt the antipsychotic treatment to the individual needs of the patient. The establishment of associations between the genetically predicted CYP2D6 metabolizer status and antipsychotic-induced weight gain (AIWG) is the first step in developing predictive tools to identify patients at high risk of AIWG.

The CYP2D6 enzyme plays an important role in the oxidative drug metabolism of several antipsychotics. The gene encoding CYP2D6 is highly polygenic. More than 100 allelic variants have been identified and assigned a phenotypic enzymatic activity score ranging from zero to two [7]. Depending on the diplotype’s total activity score, four CYP2D6 phenotypes can be predicted: poor metabolizers (PMs) who have an activity score of zero, intermediate metabolizers (IMs) who have an activity score between 0.5 and 1, normal metabolizers (NMs) who have an activity score between 1.25 and 2.25 and ultrarapid metabolizers (UMs) with an activity score greater than 2.25 [8]. While PMs and IMs have a decreased metabolic capacity compared to NMs, UMs capacity is increased. Allele and phenotype frequencies vary between ethnic groups. While CYP2D6 *4 is common among Europeans, CYP2D6*17 is more prevalent among Africans and CYP2D6*10 among Asians [9]. The frequency of predicted phenotypes varies. An average estimate for the predicted phenotype frequencies in European populations is 5.4% for PMs and 3.1% for UMs [10].

Recommendations regarding dose adjustments are typically based on these four genetically predicted phenotypes. However, their significance for the medical decision-making process varies greatly. Clozapine, for instance, is only metabolized to a lesser extent by CYP2D6. Although associated with a high AIWG potential, dose adjustment is not considered necessary in patients with a CYP2D6 genotype associated with poor metabolizer (PM) status. The same is true for olanzapine [11,12]. In comparison, altered CYP2D6 metabolizing status significantly affects the pharmacokinetics of both aripiprazole and risperidone [13], with corresponding recommendations for dose adjustment [14]. These two antipsychotics, on the other hand, are associated with a low to moderate risk of AIWG [15].

Studies suggest that altered CYP2D6 metabolizer status is associated with the risk of AIWG. Jallaq and colleagues conducted a retrospective review of electronic medical record data of children and adolescents with a mood disorder receiving oral aripiprazole. They found CYP2D6 PMs to be more likely to have weight gain measured as changes in body mass index (BMI) percentiles compared to CYP2D6 normal metabolizers (NMs) [16]. In children and adolescents diagnosed within the autistic spectrum and treated with risperidone for at least one year, Correia and colleagues found lower BMI and waist circumference in CYP2D6 ultrarapid metabolizers (UMs) compared to NMs, but no differences between PMs and NMs [17]. Ellingrod and colleagues found a significant association between weight gain and carriers of one non-function CYP2D6 allele in 11 patients treated with olanzapine [18].

Still, small samples, inconsistencies in the outcome and a mismatch between the weight-inducing effect and CYP2D6 dependence make it difficult to recommend prospective adjustments to the medical antipsychotic treatment aimed at preventing AIWG.

In a recently published randomized controlled trial (RCT), we investigated the clinical utility of routine genotyping of CYP2D6 and CYP2C19 in patients with schizophrenia [19]. Our primary outcome was antipsychotic drug persistence as an overall measure of drug effectiveness and tolerability. Secondary outcomes were positive psychotic and adverse drug reactions using the SAPS and UKU scales [20,21]. However, we also collected data on body weight and body mass index (BMI), as well as data quantifying the antipsychotic drug treatment during the entire study period.

Since CYP2D6 is more important for the metabolism of antipsychotics than CYP2C19, and CYP2D6 PMs make up the majority of the extreme metabolizers included in the original study, we have chosen in this secondary study to focus on the genetically predicted CYP2D6 metabolism.

Thus, the purpose of this study was to estimate whether poor metabolizer status for CYP2D6 is associated with AIWG and whether CYP2D6 genotype-guided drug treatment is associated with reduced weight gain.

## 2. Material and Methods

### 2.1. Randomized Controlled Trial

The design and methods of RCT were described in detail earlier in the paper [19]. Briefly, the RCT used an enriched design to increase the proportion of participants exerting poor or ultra-rapid metabolism for CYP2D6 or CYP2C19 (extreme metabolizers (ExM)). Patients were eligible for inclusion when they were at least 18 years old, diagnosed within the schizophrenic spectrum (ICD-10 codes, F20-F29) and had not been previously CYP tested. Six hundred patients were screened, and 300 NMs were randomly excluded. The remaining 240 NMs and 60 ExMs were randomly allocated to one of three study arms. In the CYP Test Guided arm (CTG), antipsychotic drug treatment was guided by an openly available CYP test. During Structured Clinical Monitoring (SCM), arm treatment was guided using a structured questionnaire to assess adverse events. Standard care served as the control arm (CNT). All patients were genotyped, but the CYP test results remained concealed in the SCM and CNT arms. The study duration was one year; outcome parameters, including body weight and BMI, were measured at baseline and at the one-year follow-up. Information on antipsychotic drug treatment and dosage was collected continuously throughout the study period.

### 2.2. Study Population Included in the Present Analysis

Patients from the original RCT who had been weighed at both baseline and follow-up were included in the present analysis. Table 1 shows the distribution of the genetic predicted CYP2D6 metabolizer status in patients included in the present analysis compared with the original RCT.

### 2.3. CYP2D6 and CYP2C19 Genotyping

Samples were genotyped for CYP2D6*3, *4, *5, *6 and CYP2D6 gene duplication/multiplications at the Research Institute of Biological Psychiatry, the Mental Health Centre Sct. Hans, Roskilde, Denmark, according to previously published methods [22]. Individuals with no functional alleles were characterized as PMs, individuals with one nonfunctional allele were defined as intermediate metabolizers (IMs), individuals with two functional alleles were NMs and individuals with duplicates of functional alleles were UMs. Table 2 provides an overview of the phenotypic activity score of the analyzed CYP2D6 alleles and their frequency in a European population.

### 2.4. Chlorpromazine Equivalents and Average Daily Exposure to Antipsychotics

For each patient, we calculated the average daily exposure to antipsychotics (ADE_AP_) as:$${\mathrm{ADE}}_{\mathrm{AP}}=\frac{\sum_{k=1}^{n}t_{k}\times D_{k}\times CPZ_{eq}}{365},$$
where *t* is the duration of exposure to a given dose in days and *D* is the prescribed daily dose in mg/day. The dose of each antipsychotic was standardized to chlorpromazine equivalents (CPZEq), i.e., the lowest effective dose of an AP that equals 200 mg chlorpromazine (estimated according to previously published methods [22]).

### 2.5. Statistical Analyses

The patients’ baseline characteristics were compared using one-way analysis of variance (ANOVA), the Kruskal–Wallis test and the Chi-square test, where appropriate. Weights at baseline and at one-year follow-up were compared using a paired *t*-test.

The association between weight change and genetically predicted CYP2D6 metabolizer status (PM, IM, NM, UM) was quantified as an unadjusted estimate and by way of multivariable linear regression with the following co-variates: BMI at baseline (continuous); age (continuous); gender (male/female); disease duration (continuous); RCT arm (CNT/CTG/SCM); exposure to FGA (yes/no); exposure to SGA with a high risk of AIWG (clozapine or olanzapine; yes/no); exposure to SGA with medium risk of AIWG (quetiapine, risperidone or paliperidone; yes/no); exposure to SGA with low risk of AIWG (amisulpride, aripiprazole, sertindole, sulpiride or ziprasidone; yes/no); mean CPZeq per day (continuous); exposure to CYP2D6 depending antipsychotics (yes/no). Test assumptions were controlled by visual inspection of residual plots, QQ-plots, homoscedasticity plots, residual versus order plots and correlation matrices combined with D’Agostino–Pearson test and variance inflation factor (VIF) statistics.

Secondly, we used multivariable linear regression to model the association between the logarithm of the relative weight change (weight at follow-up/baseline weight) and genetically predicted CYP2D6 metabolizer status (PM, IM, NM, UM). The model adjusted for the following co-variates was applied: age (continuous); gender (male/female); disease duration (continuous); RCT arm (CNT/CTG/SCM); exposure to FGA (yes/no); exposure to SGA with high liability for weight gain (clozapine or olanzapine) (yes/no); exposure to SGA with medium liability for weight gain (quetiapine, risperidone or paliperidone) (yes/no); exposure to SGA with low liability for weight gain (amisulpride, aripiprazole, sertindole, sulpiride or ziprasidone) (yes/no); mean CPZeq per day (continuous); exposure to CYP2D6-dependent antipsychotics (yes/no).

*p*-values < 0.05 were considered statistically significant. We present the results as means with 95% confidence intervals (CI) or medians with 25 and 75% quartiles. We did not correct for multiple testing.

GraphPad Prism version 9.4.0 for Windows, GraphPad Software, San Diego, CA, USA (www.graphpad.com) was used for the analyses.

## 3. Results

In total, 221 (71%) of the 311 patients in the original RCT were included, 102 (46%) of whom were males. Baseline weights ranged from 47 to 118 kg for males and 59 to 160 kg for females. Furthermore, 27 patients (12%) were CYP2D6 PMs, 75 (34%) were IMs, 111 (50%) were NMs and 8 (4%) were UMs. Overall, 60 (27%) had been exposed to FGAs and 195 (88%) to SGAs. The baseline characteristics in Table 3 were well-balanced, although PMs tended to be heavier and the UMs tended to be younger with shorter durations of disease, lower body weights and less exposure to FGA. None of these differences were statistically significant.

Figure 1 shows patient weights at baseline and follow-up. At the group level, no significant changes were observed: the average weight change was 0.3 kg (95% CI: −0.7 to 1.4 kg). At the individual patient level, body weights were far from stable: 101 (46%) patients had lost 0.2 to 49 kg, 115 (52%) had gained 0.1 to 34 kg and 41 (19%) patients had gained ≥5 kg.

Table 4 shows the results of the multiple linear regressions. Both analyses indicate a clear effect of genetically predicted CYP2D6 metabolizer status on weight change with significant weight gains in both PMs and UMs, in turn almost unaffected by adjustment for co-variates.

As the D’Agostino–Pearson test indicated that the residual values were not normally distributed, we conducted a complementary test excluding four patients identified as outliers. All patients had excessive weight changes (a female NM with a 23.9 kg weight loss and three males who were NM, IM and PM, two of whom had gained 33.9 and 34.0 kg and one who had lost 49.0 kg). In this analysis, the association between weight change and PM status remained but was no longer statistically significant (2.29 kg (95% CI: −0.37 to 4.94)). In contrast, the association between weight change and UM status remained statistically significant (6.72 kg (295% CI: 0.05 to 11.4)).

In addition to the CYP2D6 genotype, both baseline BMI and CPZeq per day were statistically significantly associated with weight change in the adjusted analysis. The analysis indicated that a higher baseline BMI of 10 kg/m^2^ was associated with a 2.37 kg lower weight gain (95% CI: −4.41 to −0.34 kg). Likewise, average exposure to 100 CPZeq per day would result in a 0.41 kg higher weight gain (95% CI: 0.05 to 0.77 kg). Figure 2 illustrates the patients’ weight change by baseline BMI and CYP2D6 genotype and Figure 3 weight changes by average daily CPZeq and CYP2D6 genotype.

Table 5 shows the results of the multiple linear regression of the relative weight change. The analysis revealed a statistically significant association between UM status and weight gain and a near-significant association between PM and weight gain. CPZeq per day was the only co-variate with a statistically significant association. Reanalysis with the exclusion of three outlier patients only marginally affected these estimates and did not change the conclusion.

## 4. Discussion

Our analysis shows a significant association between AIWG and CYP2D6 PM and UM status but no association between AIWG and the study arms of the original RCT.

Antipsychotic-induced weight gain is a serious adverse effect that increases the risk of cardiac and metabolic disease and reduces life expectancy [4,5,6]. Although antipsychotics have different risk profiles, and the second-generation antipsychotics clozapine, olanzapine and quetiapine are associated with a particularly high risk of weight gain, most antipsychotics can cause weight gain [15,24]. Data also suggest that weight gain is a dose-dependent adverse effect, although the results are not unequivocal [25], making it reasonable to assume that mechanisms causing increased plasma concentration of antipsychotics (such as decreased metabolic capacity) may amplify AIWG. In particular, CYP2D6 has been the focus of clinicians and scientists, as the enzyme contributes to the metabolism of a large proportion of frequently used antipsychotics and the gene encoding CYP2D6 exhibits genetic polymorphism [26]. So far, however, the evidence supporting that reduced metabolizing capacity in CYP2D6 is associated with an increased risk of AIWG is ambiguous.

A recently published systematic review examining whether patients with CYP2D6 PM status on long-term antipsychotic drug treatment have an average higher body weight than patients with normal CYP2D6 metabolic capacity shows conflicting results [27]. This might be a consequence of the small number of PMs (N = 93, 4.5% of all patients), notwithstanding the large size and comprehensive nature of the review. In addition, the lack of access to individual data prevents correction for possible confounders such as dose levels and treatment duration. Furthermore, data in the meta-analysis correspond to cross-sectional measurements in different CYP2D6 metabolizer groups and does not assess individual patients’ weight over time. Thus, results might be limited by heterogeneity and likely confounding, as weight gain is highly multifactorial.

In this secondary study of an RCT, we took a closer look at the association between the genetically predicted metabolizing capacity of CYP2D6 and AIWG in a population of patients with schizophrenia and treated with various antipsychotics over the 12-month study period. These data are particularly interesting as they were collected prospectively in a longitudinal design and under controlled, uniform conditions, along with data that may influence weight gain, in turn enabling adjustment in regression analyses. The enriched design of the RCT ensured a significantly higher proportion of PMs (12%) than normally expected in European populations [10], increasing data utility by reducing imbalances.

Baseline data showed no statistically significant difference in body weight and BMI between the four metabolizer groups, although the UMs showed a tendency towards lower body weight and BMI. At the same time, the UMs appeared to be younger, include a higher proportion of males, and have a shorter disease duration (Table 3). Although these data are not statistically significant and their conclusiveness is limited by the small sample size, these differences might explain the apparently lower body weight and BMI in the UMs at baseline.

The multiple regression analyses showed that both CYP2D6 PM and UM status are significantly associated with increased weight gain, a finding that remains statistically significant after adjustment for co-variates (Table 4). Furthermore, we found a positive association between weight gain and the average amount of daily CPZeq (Table 4, Figure 2). Interestingly, we found no correlation between weight gain and the use of FGA or SGAs with particular weight-gain potential (Table 4). Thus, our finding supports the notion that weight gain in PMs is driven by a reduced metabolic CYP2D6 capacity and increased antipsychotic exposure in general rather than the use of antipsychotics with high weight-gain potential.

In accordance with our findings, a recent meta-analysis found that AIWG was dose-dependent, and although the propensity of different antipsychotics to trigger weight gain varied considerably, no antipsychotic was completely exempted from this [15,25].

Less obvious than the mechanisms behind CYP2D6 PM status and weight gain are the mechanisms behind the weight gain found in CYP2D6 UMs. Not only should the increased enzymatic capacity keep antipsychotic exposure down, but UMs also tend to have been treated with lower CPZeq doses than other metabolizer groups (Table 3). One explanation might be their shorter disease duration and lower age, as weight gain appears to be most rapid in the early period after commencing antipsychotic drug therapy [28] and occurs more frequently in younger patients [24]. Another possible explanation might be that UMs in this study tend to have a lower baseline BMI (Table 3), as baseline BMI appears to be negatively associated with weight gain (Table 4, Figure 3). We found no data in the literature to support this finding. As mentioned earlier, these considerations are highly speculative due to the small sample size of UMs and, at best, hypothesis-generating.

Since our data do not fully meet the prerequisites for multiple regression analysis, we chose to conduct a complementary analysis (Table 5), where we excluded four outliers. Although only marginally significant for the PMs, the results of this analysis support the results of the first analysis (Table 4).

The pharmacological treatment of mental illness is complex and rarely follows a fixed treatment algorithm. Different underlying mechanisms have been proposed for AIWG, including genetic factors that might explain different susceptibilities to the weight gain of patients treated with antipsychotics [29]. Therefore, CYP2D6 status as an isolated factor may not suffice individual guide treatment. Instead, the information should be integrated into the overall medical decision-making process, both in terms of drug selection and dosage, but also contribute to the understanding of the patient’s individual treatment response.

Originally, the RCT was designed to investigate whether awareness of the CYP2D6 and CYP2C19 genotypes would influence prescribing patterns and the resultant tolerability of the given antipsychotic treatment. Thus, one could expect equal or less weight gain in the CTG study arm compared to the SCM and CNT arm as the attending physician was given the opportunity to adapt antipsychotic drug selection and dosage to the patients’ CYP2D6 metabolizing capacity. However, our results suggest no such effect (Figure 1, Table 4); that is, awareness of CYP2D6 status (to guide treatment adjustments) does not seem to prevent or reduce weight gain. One explanation may be that most CYP2D6-dependent antipsychotics are not or only to a lesser extent associated with weight gain, whereas several antipsychotics (including clozapine, olanzapine and quetiapine) whose metabolism does not or only to a lesser extent depend on CYP2D6 have a significant weight gain-inducing effect (including clozapine, olanzapine and quetiapine). Thus, switching antipsychotics based on patients’ CYP status is not an obvious choice if one intended to prevent weight gain.

### Strengths and Limitations

This study has several strengths. First, it used data collected prospectively in a well-conducted RCT. Second, considering its detailed phenotypic data, the study population was relatively large. Third, the enriched design of the original RCTs mitigated the imbalance often seen in observational studies in the domain of antipsychotics, metabolizing status and weight gain. Fourth, seeking to answer the same question with two kinds of regression models (modeling the absolute weight change in one and the relative change in the other) and observing qualitative agreement is reassuring. Fifth, the linear regression models yield simple estimates of strengths of associations, useful for association studies such as this; if the purpose were patient-level prediction, these would serve as good starting points, perhaps extended with splines for non-linear effects.

However, the study also has some limitations. First, unfortunately, only 71% of the original RCT study population could be included in this study (due to missing data on the outcome variable). The summary characteristics, however, did not raise suspicion of missingness-not-at-random, in turn suggesting that no selection bias was introduced. Second, although the proportion of extreme metabolizers is relatively high in this study, the low absolute number of extreme metabolizers does hurt the estimation of main and co-variate effects. Third, standardizing antipsychotic exposure to daily CPZeq is an approximation that will not perfectly reflect the real exposure level. It is, however, widely used and helps keep the regression models as parsimonious as possible. Fourth, as essentially all studies using drug dosing information, we assume that the prescribed doses are the same as those administered and ingested. While this assumption is unverifiable, we expect deviations to be independent of the CYP2D6 status and, thus, to not bias the estimates.

## 5. Conclusions

Our results show that the genetically predicted CYP2D6 phenotype matters for AIWG. Although one would not recommend switching a patient with CYP2D6 PM status to a CYP2D6-independent antipsychotic with greater weight-gain potential, weight gain might be limited or completely avoided if doses were consistently adjusted downward to reach minimum effective doses. This would be an interesting avenue to explore further, and an RCT seems a well-suited vehicle to this end.

## Figures and Tables

**Figure 1 jpm-12-01728-f001:**
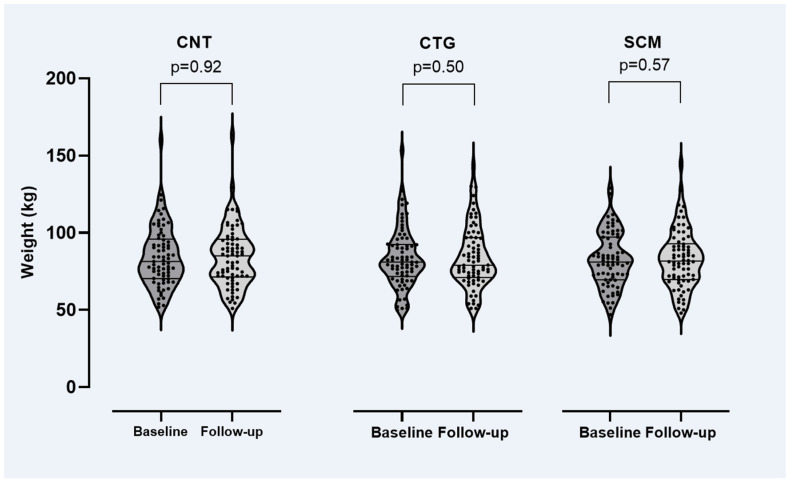
Weight by study arm at baseline and follow-up of the RCT (see main text for details). CNT = control study arm, CTG = genotype-guided study arm, SCM = structured clinical monitoring arm. Paired t-test was used for the comparisons.

**Figure 2 jpm-12-01728-f002:**
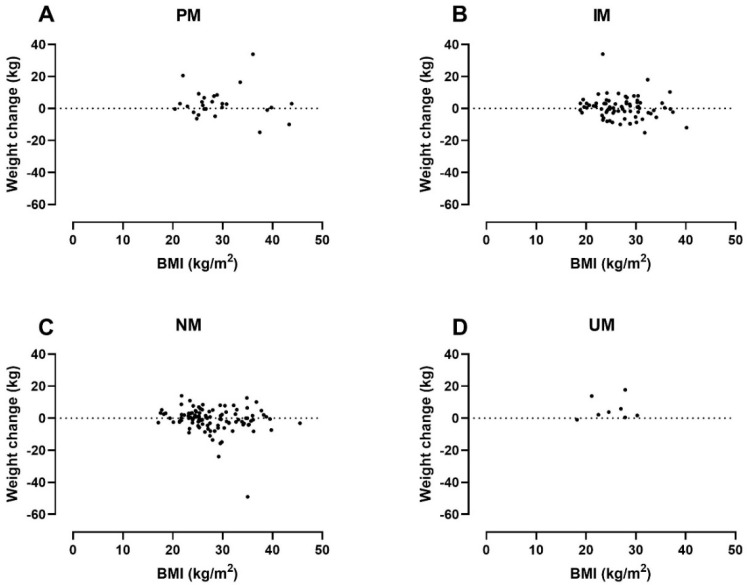
Weight change by baseline body mass index (BMI) and CYP2D6 genotyp in: PMs (=Poor metabolizers) (**A**); IMs (=Intermediate metabolizers) (**B**); NMs (=Normal metabolizers) (**C**) and UMs (=Ultrarapid metabolizers) (**D**).

**Figure 3 jpm-12-01728-f003:**
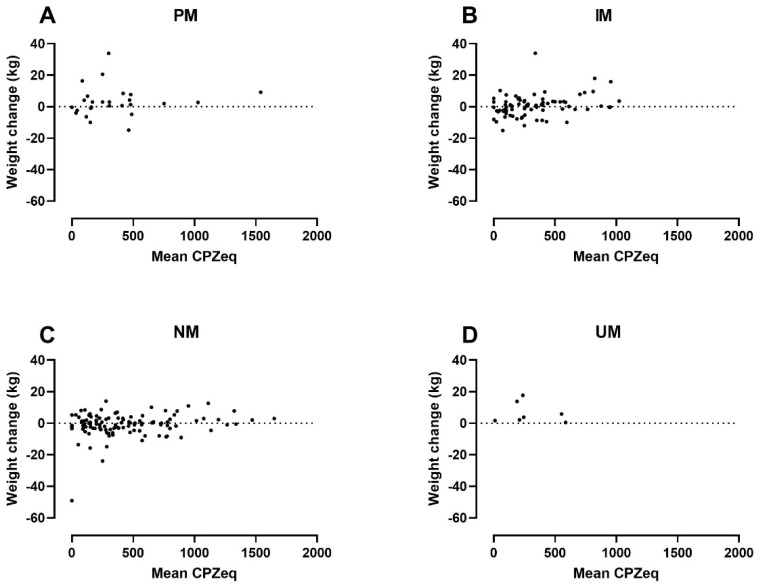
Weight change by average daily exposure to antipsychotics and CYP2D6 genotype in: PMs (=Poor metabolizers) (**A**); IMs (=Intermediate metabolizers) (**B**); NMs (=Normal metabolizers) (**C**) and; UMs (=Ultrarapid metabolizers) (**D**). CPZeq = Chlorpromazine equivalents.

**Table 1 jpm-12-01728-t001:** Comparison of the number of patients and CYP2D6 metabolizer status included in the original RCT and the present analysis.

RCT Study Arm	N in RCT	Excluded Due to Missing Weight Data	N in Present Analysis
CTG	Total 95	Total 21	Total 74
	PM 13	PM 5	PM 8
	IM 30	IM 4	IM 26
	NM 47	NM 11	NM 36
	UM 5	UM 1	UM 4
SCM	Total 94	Total 20	Total 74
	PM 12	PM 5	PM 7
	IM 32	IM 7	IM 25
	NM 46	NM 7	NM 39
	UM 4	UM 1	UM 3
CNT	Total 101	Total 28	Total 73
	PM 15	PM 3	PM 12
	IM 33	IM 7	IM 24
	NM 50	NM 14	NM 36
	UM 3	UM 2	UM 1

N = Number of patients, CTG = CYP Test Guided study arm, SCM = Structured Clinical Monitoring study arm, CNT = Control study arm.

**Table 2 jpm-12-01728-t002:** Activity score for analyzed CYP2D6 alleles and their frequencies in Europeans [23].

CYP2D6 Allele	Activity Score	Phenotypic Activity Score	Allele Frequencies i Europeans
CYP2D6*3	0	Zero function	1.60
CYP2D6*4	0	Zero function	18.50
CYP2D6*5	0	Zero function	3–5
CYP2D6*6	0	Zero function	0.5
Duplications	*	*	NA

* Activity score depends on the allele.

**Table 3 jpm-12-01728-t003:** Demography.

	CYP2D6 PM	CYP2D6 IM	CYP2D6 NM	CYP2D6 UM
	N = 27	N = 75	N = 111	N = 8
Mean age (years)	41.7 (37.2; 46.2)	41.5 (39.0; 44.0)	40.7 (38.8; 42.6)	34.1 (27.7; 40.4)
Male (No (%))	12 (44.4)	34 (45.3)	52 (46.8)	4 (50.0)
Median disease duration (years)	6.3 (1.5; 11.5)	6.3 (2.6; 11.3)	6.3 (3.3; 14.7)	0.9 (0.6; 8.4)
Mean weight at baseline (kg)	91.0 (81.1; 101.0)	81.4 (77.4; 85.5)	83.1 (79.8; 86.4)	77.6 (64.2; 91.0)
Mean BMI at baseline (kg/m^2^)	29.4 (26.8; 32.0)	27.2 (26.1; 28.3)	28.0 (26.9; 29.0)	24.9 (21.5; 28.3)
Mean chlorpromazine equivalents per day	342 (211; 474)	335 (274; 397)	440 (373; 507)	307 (75; 539)
Use of first-generation antipsychotics (No (%))	9 (33.3)	21 (28.0)	29 (26.1)	1 (12.5)
Use of clozapine or olanzapine (No (%))	10 (37.0)	27 (36.0)	41 (36.9)	2 (25.0)
Use of quetiapine, risperidone or paliperidone (No (%))	9 (33.3)	25 (33.3)	47 (42.3)	3 (37.5)
Use of other second-generation antipsychotics (No (%))	11 (40.7)	30 (40.0)	47 (42.3)	5 (62.5)

**Table 4 jpm-12-01728-t004:** Multiple linear regression. Average weight change.

Parameter	Estimate (95% CI)
Unadjusted	
Intercept	−0.92 (−2.3; 0.50)
CYP2D6 status (ref: NM)	
PM	4.00 (0.80; 7.21)
IM	1.58 (−0.65; 3.81)
UM	6.50 (1.03; 12.0)
Adjusted	
Intercept	−0.38 (−6.72; 7.49)
CYP2D6 status (ref: NM)	
PM	4.26 (0.88; 7.64)
IM	1.51 (−0.85; 3.87)
UM	7.26 (1.24; 13.3)
Baseline BMI (kg/m^2^)	−0.24 (−0.44; −0.03)
Age (years)	0.07 (−0.05; 0.19)
Disease duration (years)	0.007 (−0.15; 0.16)
RCT arm (ref: CNT)	
CTG	0.30 (−2.28; 2.87)
SCM	0.57 (−2.02; 3.16)
Exposed to FGA	−1.05 (−3.94; 1.83)
SGA high	−0.48 (−3.10; 2.15)
SGA medium	0.61 (−1.92; 3.13)
SGA low	0.70 (−1.78; 3.19)
Mean daily CPZeq	0.0041 (0.0005; 0.0077)
CYPA2D6-dependent drugs	1.12 (−1.43; 3.67)
Male sex	−0.90 (−3.06; 1.26)

BMI = Body mass index; RCT = Randomized controlled trial; CNT = control study arm; CTG = CYP-test guided study arm; SCM = Structured clinical monitoring study arm; FGA = First generation antipsychotics; SGA high/medium/low = Second generation antipsychotics with high/medium/low liability for weight gain; CPZeq = Chlorpromazine equivalents; CYP2D6 = Cytochrome P450 2D6.

**Table 5 jpm-12-01728-t005:** Multiple linear regression and relative weight change.

Model	Intercept (kg)	CYP2D6	Estimate (kg)	*p*-Value
Adjusted	0.99 (0.97;1.01)	Normal metabolizer	Ref.	-
		Poor metabolizer	1.01 (1.00;1.03)	0.070
		Intermediate metabolizer	1.00 (0.99;1.01)	0.43
		Ultra-rapid metabolizer	1.04 (1.01;1.06)	0.027

## Data Availability

Deidentified individual-participant data that underlie the results presented in this Article, the study protocol, statistical analysis plan, and analytical code will be available to investigators for individual participant data meta-analyses that have been approved by independent review committees. Data will be available from the publication date of this Article, with no end date. Proposals for use of data and requests for access should be directed to gju@regionsjaelland.dk. There can be restrictions and conditions for data sharing according to GDPR.

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
