# Peer review of "Is the CYP2D6 Genotype Associated with Antipsychotic-Induced Weight Gain?†"

_jpm, 2022, doi:10.3390/jpm12101728_

Round 1

Reviewer 1 Report

Specific comment:

 Nowadays, personalized medicine is fruitful branch of knowledge not only promising for patients better live but also helps clinician to better treat and predict diagnosis. Schizophrenia affects 0.32% of human population with onset during late adolescence. As like major depressive disorders pathomechanism of disease is not understandable and treatment is base on medication and psychosocial therapy. However, antipsychotics used in schizophrenia can caused metabolic disturbances with weight gain, what can tremendous affect in quality of life, especially for younger group of patients. For these reason, it is important to understand how drugs can afflicted to individuals and if can be predicted using to available methods.

Proposed study can deepen knowledge about risk of antipsychotic induced weight gain in patients as well contribute to improve treatment protocol introduced to clinic.

Abstract:

Abstract is readable and consistent.

Introduction:

Introduction seems to be correctly written. Nevertheless, could you explain briefly/ define normal, ultrarapid, low etc. CYPD6 metabolizers. --- % in the population

Materials and Method:

Add more information how you chose participant involved in your study… --- figure or table will be kindly requested --- you citied previous paper where these data are nicely presented but it can be necessary for readers to have description also in this manuscript.

The same is for genotyping --- sum the information on table/figure

Results and Discussion:

Correctly and readable.

Author Response

Thank you for your constructive comments, which we have complied with as follows:

  • Introduction: Introduction seems to be correctly written. Nevertheless, could you explain briefly/ define normal, ultrarapid, low etc. CYPD6 metabolizers. --- % in the population

We have included a corresponding statement in the introduction (Introduction, section 4, page 2-3).

  • Materials and Method: Add more information how you chose participant involved in your study… --- figure or table will be kindly requested --- you citied previous paper where these data are nicely presented but it can be necessary for readers to have description also in this manuscript.

We have included inclusion criteria for the RCT in the method section (Methods, Randomized Controlled Trial, section 1, page 4)

Furthermore, we have included an additional table that describes how many patients from each study of the original RCT are included in the present analysis divided by CYP2D6 phenotypes (table 1, page 4-5)

  • The same is for genotyping --- sum the information on table/figure

We have included an additional table in the method section that describes activity score and allele frequencies in Europeans (table 2, page 5).

We have also carried out a linguistic revision to correct errors and improve readability.

Sincerely,

Gesche Jürgens

Reviewer 2 Report

  • A brief summary 

The present article titled “Is CYP2D6 genotype guided treatment associated with reduced antipsychotic induced weight gain?is focused on the studies proving the link between altered CYP2D6 metabolizer status and the risk of antipsychotic induced weight gain and using it as predictive tool.

  • Comments 

Introduction

The introduction is comprehensive and highlights the role of CYP2D6 enzyme in metabolism of some antipsychotics and proves the importance to be genetically predicted the CYP2D6 metabolizer status in order to be identified patients at high risk of antipsychotic induced weight gain. 

Materials and methods

-          To the description of the group tested, please add information about age; gender; disease duration.

-          - Were any other blood tests (eg, glucose, cholesterol levels) performed on the patients in the group that may be associated with some pathological consequences of weight gain?

The paper can be published after minor revision.

Author Response

Thank you for reviewing our manuscript. We have addressed your questions point by point below.

  • To the description of the group tested, please add information about age; gender; disease duration.

Age gender and disease duration are included in table 3, page 6-7.

  • Were any other blood tests (eg, glucose, cholesterol levels) performed on the patients in the group that may be associated with some pathological consequences of weight gain?

Unfortunately, we do not have additional parameters that could be associated with metabolic complications.

We have made minor changes to the manuscript as part of a linguistic revision to correct errors and improve readability.

Sincerely,

Gesche Jürgens